# Genomic Characterization of a Relative of Mumps Virus in Lesser Dawn Bats of Southeast Asia

**DOI:** 10.3390/v15030659

**Published:** 2023-02-28

**Authors:** Adrian C. Paskey, Xiao Fang Lim, Justin H. J. Ng, Gregory K. Rice, Wan Ni Chia, Casandra W. Philipson, Randy Foo, Regina Z. Cer, Kyle A. Long, Matthew R. Lueder, Lindsay Glang, Kenneth G. Frey, Theron Hamilton, Ian H. Mendenhall, Gavin J. Smith, Danielle E. Anderson, Lin-Fa Wang, Kimberly A. Bishop-Lilly

**Affiliations:** 1Genomics and Bioinformatics Department, Biological Defense Research Directorate, Naval Medical Research Center-Frederick, Fort Detrick, Frederick, MD 21702, USA; 2Leidos, Reston, VA 20190, USA; 3Programme in Emerging Infectious Diseases, Duke-NUS Medical School, Singapore 169857, Singapore; 4Defense Threat Reduction Agency, Fort Belvoir, VA 22060, USA; 5Victorian Infectious Diseases Reference Laboratory, The Peter Doherty Institute for Infection and Immunity, Melbourne, VIC 3000, Australia

**Keywords:** paramyxoviruses, *Eonycteris spelaea*, next-generation sequencing, genomics, virus discovery

## Abstract

The importance of genomic surveillance on emerging diseases continues to be highlighted with the ongoing SARS-CoV-2 pandemic. Here, we present an analysis of a new bat-borne mumps virus (MuV) in a captive colony of lesser dawn bats (*Eonycteris spelaea*). This report describes an investigation of MuV-specific data originally collected as part of a longitudinal virome study of apparently healthy, captive lesser dawn bats in Southeast Asia (BioProject ID PRJNA561193) which was the first report of a MuV-like virus, named dawn bat paramyxovirus (DbPV), in bats outside of Africa. More in-depth analysis of these original RNA sequences in the current report reveals that the new DbPV genome shares only 86% amino acid identity with the RNA-dependent RNA polymerase of its closest relative, the African bat-borne mumps virus (AbMuV). While there is no obvious immediate cause for concern, it is important to continue investigating and monitoring bat-borne MuVs to determine the risk of human infection.

## 1. Introduction

Paramyxoviruses have negative-sense, single-stranded RNA genomes and include notable human pathogens such as mumps virus and measles virus [1]. Bats are known hosts of paramyxoviruses, such as henipaviruses, which are deadly to humans but cause no apparent disease in bats [1]. As of October 2022, there are 295 entries for bat paramyxoviruses in DBatVir, the database of bat-associated viruses, six of which are associated with lesser dawn bats (*Eonycteris spelaea*) [2]. Unclassifiable genomic sequences related to paramyxovirus large (*L*) and nucleoprotein (*NP*) genes were previously reported in fecal samples collected from lesser dawn bats in Singapore [3]. Additionally, the L gene (KC599257) of a henipa-related virus was detected in lesser dawn bat fecal samples in China [4]. To our knowledge, no full-length paramyxovirus genomes detected in lesser dawn bats have been published. In the current report, we present a deeper follow-up to our study of the virome of lesser dawn bats (*Eonycteris spelaea,* a highly gregarious species known to co-roost in large populations of multiple bat species in caves across Southeast Asia [5]), in the form of a genetic characterization study of a novel MuV using data originally collected as part of that virome characterization effort, as well as our efforts to isolate this new bat-borne mumps virus, named dawn bat paramyxovirus (DbPV).

Human mumps virus (MuV) causes a vaccine-preventable childhood disease that is predominately associated with fever and parotid swelling. Prior to routine administration of the measles, mumps, and rubella (MMR) virus vaccine to children, MuV was the leading cause of encephalitis in the United States [6]. Although there are twelve genotypes of human MuV, one possible reason for the great success with vaccinations is that MuV is thought to only have one serotype [7]. A study of the antigenic relatedness between the human mumps virus and the related African bat-borne mumps viruses (AbMuV), of the genus *Rubulavirus*, suggested that they also belong to one serogroup [1]. In fact, the shared amino acid (aa) identity between AbMuV (HQ660095) and MuV is above 90% among nucleoprotein (NP), matrix (M), and large (L) protein sequences. Bats are ancient mammals, having had ample time to coevolve with viruses [8], and bat-borne viruses continue to emerge from the wild, as evidenced by the COVID-19 pandemic [9]. As we continue to combat ancient diseases such as MuV, which may have been first described by Hippocrates [6], it is important to catalog and investigate newly detected relatives of viruses of pandemic potential. This report describes our genetic characterization of MuV-derived sequencing reads detected previously in a longitudinal virome study of an apparently healthy, captive lesser dawn bat colony in Southeast Asia (BioProject ID PRJNA561193) [10] and our attempts to isolate this virus from noninvasively collected environmental samples.

## 2. Materials and Methods

### 2.1. Library Preparation and Sequencing

Sequencing reads used for this study were generated from a longitudinal virome study of a captive lesser dawn bat colony in Singapore [10], with details of the breeding colony published by Foo et al. [11]. In brief, head, body, oral, and rectal swabs were collected for sequencing from each individual bat in the colony across six time points spanning 18 months from April 2016 through September 2018. Sequencing was performed on the NextSeq 500 platform using V2 chemistry with 2 × 150 base pair (bp) read lengths.

### 2.2. Scaffolded Genome and Read Mapping to Generate a Composite Reference

Methods were previously published, and the reads were deposited to BioProject ID PRJNA561193 [10]. The majority of the reads contributing to the composite genome MT506196 originated from a body swab collected in April 2016 from bat 7633EDB; however, reads from other samples were used to fill in gaps as much as possible. First, a draft scaffold genome was generated solely from the reads derived from a single swab (from bat 7633EDB) that were assembled using metaSPAdes [12] and, in conjunction with stitched reads identified using VirusSeeker [13], were mapped with low stringency to a near neighbor, AbMuV (HQ660095). The resulting consensus sequence was extracted with gaps filled in as “Ns” using CLC Genomics Workbench V11 (QIAGEN Bioinformatics; Redwood City, CA, USA) to create a scaffolded consensus genome. To fill in gaps, stitched reads from VirusSeeker and contigs assembled using metaSPAdes from other samples, as well as trimmed reads from the original swab from bat 7633EDB, were mapped iteratively with increasing stringency (requiring half of the read to match the reference with a minimum of 65% identity, iteratively increasing to 90%) to the scaffolded reference to “build out” the existing regions of coverage where possible. The resulting consensus was used as the scaffolded, composite genome representing reads from multiple samples and filled with “Ns” where sufficient coverage did not exist in the dataset. The reads from bat 7633EDB covered ~75% of the expected length of the scaffolded composite genome as shown in Figure 1.

Read mapping to understand colony-level prevalence (results summarized in Table 1) was performed using default settings in bbmap [14] with a reference index consisting of the scaffolded DbPV reference that was generated as described above. For a particular swab to count as DbPV-positive, at least 10 reads were required to map with 95% nucleotide identity across each read. Bbmap uses an algorithm that determines mapping agreement based on the ratio of each read’s alignment score to the maximum alignment core (where 100% of bases match the reference).

### 2.3. Phylogenetic Analysis

Maximum likelihood trees were generated using the complete SH gene product (Figure 2) and the partially complete L gene product (Appendix A) from bat and human MuV sequences. The reference sequences selected for phylogenetic analysis of SH represent all human genotypes in addition to bat-borne MuVs. Alignments were generated using CLC Genomics Workbench. IQ-TREE v2.0.3 [15,16] was used to generate a consensus tree with an mtVer+I model of substitution (log-likelihood-755.83) from 1000 bootstrap trees, and FigTree v1.4.4 was used for visualization [17]. Bootstrap values and branch lengths are labeled; the tree is unrooted.

### 2.4. Comparison of Amino Acid Identities among Mumps Viruses

To evaluate the minimum identity among all MuVs, all available amino acid sequences were downloaded from the Virus Pathogen Resource (ViPR) database [18]. Comparison of MuV strains was performed by calculating the percent identity among amino acid sequences of nucleoprotein (NP), V and phosphoprotein (P/V), matrix (M), fusion (F), small hydrophobic (SH), hemagglutinin (HN), and large (full-length L, as well as a subset of RNA-dependent RNA polymerase sequence) of Human mumps virus (MuV, NC_002200), African bat mumps virus (AbMuV, HQ660095), and the Dawn bat paramyxovirus (DbPV, MT506196) using the pairwise comparison function for a multiple sequence alignment in CLC Workbench.

## 3. Results

### 3.1. Similarity among Novel and Previously Known Mumps Virus Genomes

Previously, we reported a longitudinal study of the virome in a captive lesser dawn bat colony between 2016 and 2018, where we noted some evidence of paramyxoviruses, among other viruses [10]. In the current study, we sought to characterize the novel MuV, and so we performed an in-depth analysis of data from 114 of these swabs from which sequences potentially belonging to a novel mumps virus (which we now refer to as Dawn bat paramyxovirus or DbPV) were detected. DbPV shares 86% identity at the amino acid level to the published bat MuV reference (HQ660095), and to human MuV (NC_002200) RNA-dependent RNA polymerase (RdRp). We found that contigs from one body swab collected from bat 7633EDB in April 2016 cover 76.5% of the expected length of the DbPV genome with an average coverage depth of 9.6x (Figure 1). The genome architecture of the human and bat mumps viruses is estimated to be similar in organization, as indicated by top BlastX results demonstrating similarity among MuV nucleoprotein (NP), matrix (M), fusion (F), hemagglutinin (HN), and large (L) genes for the contigs that comprise the composite genome for DbPV (Appendix A); however, due to incompleteness of the genome, the possibility of a previously unknown gene (analogous to the finding of X genes in the Beilong paramyxovirus) may exist [19]. Comparison of the SH sequence of DbPV to previously known sequences demonstrates that it is not more closely related to the bat virus than to human MuV [1] sequences (Figure 2).

The small hydrophobic (SH) protein is typically the most divergent amino acid sequence among human MuVs [20], and that was also demonstrated in a pairwise identity analysis, with a minimum amino acid identity of 66.7% (Appendix A). While we could not calculate the amino acid identity for each amino acid sequence in its entirety due to incompleteness of the genome, we observed that DbPV shares similar identity with AbMuV and MuV (navy and maroon bars, Appendix A). While both DbPV and AbMuV were detected in bats, we observed that the DbPV is a distinct and novel virus that is similarly divergent from both the AbMuV and the MuV.

Given that proteins F and HN are the major targets of neutralizing antibodies in both AbMuV and human MuV [21,22], we sought to compare sequences for known epitopes to the analogous genes encoded in the genome of DbPV. Although the composite DbPV genome has what might be only a partial sequence for both F and HN genes, the consensus covered both the beginning and end of the predicted open reading frames (ORFs) when compared via alignment to near neighbors. As such, Ns were filled where either coverage did not exist or the novel virus genome may be shorter as compared to known near neighbors, and then ORFs were predicted using CLC Workbench and subsequently translated to the amino acid (aa) level. As shown in Appendix A where the predicted DbPV aa sequences are aligned with the corresponding aa sequences of human MuV and AbMuV, there are 30 and 139 aa long gaps in the predicted 538 aa sequence for F protein, and the predicted 587 aa sequence for HN has three estimated gaps of 64, 10, and 13 aa. Due to the short read sequencing employed, which produced 150 bp read lengths that are much shorter than the total length of either a typical paramyxovirus F or HN gene, and since there were no functional protein domains missing from the DbPV sequences, it could not be evaluated whether these apparent gaps are indicative of an incomplete sequence with one or more portions that were not sampled by reads or whether DbPV naturally encodes shorter F and HN genes. Similarly, the overall degree of aa conservation was not calculated due to potentially incomplete coverage of both the F and HN genes (see mapping in Figure 1); however, we did observe that there was a lack of conservation among the amino acid positions 323 and 373 (EU370207) in the F protein [21]. These positions are significant as they are part of at least one neutralizing conformational epitope, as determined by Šantak et al., through experiments that identified previously unknown epitopes [21].

Continuing our evaluation of the DbPV aa sequences in comparison to those of known relatives, we identified the conservation of an important cleavage motif in F. AbMuV and DbPV encode a putative multibasic furin cleavage motif (RRRKR|F) that is conserved in the two bat viruses as compared to MuV F (Figure 3) [23,24]. This is a significant finding because bat and human MuV both require the expression of sialic acid on the surface of target cells and proteolytic cleavage of the F protein by furin for host cell binding and entry. In in vitro studies of a rescued recombinant AbMuV in furin-deficient LoVo epithelial cells, Krüger et al. did not identify proteolytic activation nor did they observe a spread of infection by cell-to-cell fusion up to four days post infection under those conditions. These data implicate the significance of this cleavage site for infection by bat-borne MuVs [24]. The conservation of this cleavage site in DbPV leads us to hypothesize that the same cleavage motif would be essential for cell binding and entry of the novel virus.

### 3.2. Frequency of Detection in the Captive Colony

While the reads and contigs from one body swab collected from bat 7633EDB in April 2016 covered 76.5% of the DbPV scaffolded composite genome with an average depth of 9.6x (Figure 1), this sample is unique in that no read set from any other single sample covered more than 5% of the scaffolded reference genome. The prevalence of this virus at each time point is summarized in Table 1, as determined by read mapping, requiring 10 or more reads to map with greater than 95% identity across each read in order to call a given sample positive. Overall, we observed a limited breadth of coverage across the genome for each of these samples, and the consensus from each was overall identical to the scaffolded composite genome, with any observed single nucleotide variant occurring with less than 35% frequency and therefore not considered to be significant.

Figure 4 demonstrates the frequency of colony-level detection among swab types and individual bats. DbPV was detected most frequently in oral and rectal swabs (Figure 4A) and in samples from all bats in the colony collected in both April and October 2016 (Figure 4B). Based on these data, DbPV did not persist within a single bat for each of the six time points. It is unknown whether viral RNA copies were below the limit of detection or whether the virus was cleared by most bats by the January and May 2017 time points. The variation in which swabs were found to include evidence of DbPV appears to shift by swab type and number of swabs over time, where, in April 2016, 1 each of body and head swabs but 16 oral and 8 rectal swabs were considered positive for DbPV. In July 2016, no body or head swabs but 14 oral and 7 rectal swabs produced reads mapping to DbPV. In October 2016, 7 body, 9 head, 19 oral, and 18 rectal swabs were considered positive for DbPV. In January 2017, reads were detected in just one body and one rectal swab, and in September 2017, reads were detected in five oral and six rectal swabs. We observe that while the virus did persist in the colony over time, it was detected less frequently in swabs sequenced from later collection dates.

### 3.3. Efforts to Isolate Dawn Bat Paramyxovirus

Once the composite genome of the DbPV was identified and it was clear that most MuV-like reads were detected in a swab from bat 7633EDB, isolation of the DbPV was attempted in PaKi (*Pteropus alecto* kidney) cells. The urine sample was collected 18 months following the collection of the last DbPV-positive swab from bat 7633EDB. In brief, 5 µL of urine collected in March 2019 from Bat 7633EDB was inoculated onto 1 × 10^6^ cells in 500 µL of Dulbecco’s Modified Eagle Medium (DMEM, 1% penicillin/streptomycin) in a 6-well plate. The cells were incubated at 37 °C with 5% CO_2_ for 2 h. Following incubation, the inoculum was replaced with 1 mL of DMEM (5% FBS, 1% penicillin/streptomycin). The cells were incubated at 37 °C with 5% CO_2_ and observed daily for the presence of a cytopathic effect (CPE). Supernatant from PaKi cells was serially passaged three times. Although CPE was observed in PaKi cells in passage 3, paramyxovirus sequences could not be detected by PCR. Due to the exhaustion of primary samples from 7633EDB, we were not able to ascertain whether another bat cell type might have been more permissive for DbPV, and we were unable to isolate this virus.

## 4. Discussion

Herein we report the genetic characterization and efforts to isolate a novel bat mumps-like virus, DbPV, in lesser dawn bats. We detected DbPV viral RNA in 114 noninvasive, environmental swabs collected as part of a longitudinal study of a captive colony of lesser dawn bats. Our findings are consistent with recently published reports that paramyxoviruses persist in bat colonies, but there seems to be notable variation in detection among sampling time points [25,26]. Furthermore, a similar report described a negative RT-PCR of culture supernatant and cell lysates following inoculation with samples known to be positive for a paramyxovirus [19]. Overall, DbPV shares only 86% amino acid identity with the RdRp of AbMuV, the closest relative. The International Committee for the Taxonomy of Viruses (ICTV) rules for paramyxovirus demarcation rely upon the comparison of full-length L protein amino acid sequences [27]. The demarcation for a new species within *Paramyxoviridae* is defined as 0.03 in trees generated according to their criteria; this has been completed, and a proposal for DbPV as a new species is in preparation for submission to the ICTV (tree included in Appendix A).

To better understand how this new virus compares to other MuVs, we compared all predicted coding sequences. With the exception of SH, the DbPV coding sequences contained gaps, and thus interpretation is limited. The most divergent gene for which the complete sequence is known in DbPV is SH; this is consistent with the divergence in SH among other MuVs (as shown in Figure 2 and Appendix A), and because SH is the most divergent, it is used to define the genotypes of the human MuV [28]. For this reason, some publications compare SH gene sequences only [29], though not all orthorubulaviruses have the SH gene [30,31]. The SH protein is expressed in infected cells but not incorporated in the virion: it is thought to block the TNFα-mediated apoptosis pathway to evade the host antiviral response [32]. The function of the SH protein likely results in mutations under host pressure and could explain the high variance, and it has been described as a luxury gene due to the virus’s tolerance of its high mutation rate [20]. Across all coding sequences, DbPV has generally lower amino acid identity to both AbMuV and MuV and, based on our limited comparisons among partial sequences, is not notably more similar to one as compared to the other (Appendix A and Appendix A). Gaps in the consensus sequence of DbPV and inability to isolate the virus in PaKi cells prevent truly informative comparisons among putative epitopes based on knowledge of targets of neutralizing antibodies in AbMuV and MuV. However, there was sufficient coverage to identify the putative multibasic furin cleavage motif that may be essential for the process of cell binding and entry (Figure 3).

While the majority of MuV-like reads spanning the breadth of the DbPV composite genome were derived from a single sample, we observed that there were many samples sequenced over the course of this longitudinal study in which we found evidence of DbPV. It is not clear why the number of reads that were mapped from most samples was low, in particular as compared to the body swab from bat 7633EDB that had reads covering ~75% of the composite genome. As this composite genome was elucidated by analyzing interesting MuV-like reads following the conclusion of the collection of environmental swabs and depletion of the original biomaterial, additional attempts to complete the virus genome were not made (i.e., using RT-PCR to fill gaps). This is consistent with similar studies that report the detection of new bat viruses that were characterized as much as possible but were not finished genomes due to sample limitations [33]. We hypothesize that this newly detected virus, like other bat paramyxoviruses, may persist within individual bats without being consistently shed at detectable titers. In a review that summarized the ability to detect the longitudinal shedding of viral RNA by bats in multiple studies, it was observed that the detection of low-prevalence viruses was inconsistent, even with large sample sizes [34]. While we observed temporal variation (Figure 4), the virus was detected less frequently overall in the second half of this longitudinal study.

While there is no obvious evidence that the detection of a new bat-borne MuV is an immediate public health concern, it is important to continue to investigate bat-borne MuVs to determine the risk of human infection. By characterizing new viruses that share common ancestors to human pathogens through the lens of One Health, we gain a greater understanding of the differences among viruses that have evolved to become pathogens and those that pose no threat to humans [35]. Our discovery of DbPV in Southeast Asia provides further evidence of the ecological relationship between bats and mammalian paramyxoviruses, although a bat mumps rubulavirus yet remains to be isolated. Attempts to isolate this virus from the captive colony of lesser dawn bats were unsuccessful. In light of patterns observed by tracking COVID-19 patients who were positive by RT-PCR for SARS-CoV-2 for as long as 28 days, but from whom the virus could not be isolated beyond 8 days following the onset of symptoms, it seems this lack of virus isolation is not unusual [36]. The sample with the most reads derived from this virus was a body swab, making it difficult to determine whether the virus was shed by bat 7633EDB itself or another bat in close proximity. The gregarious nature and grooming behavior of lesser dawn bats could explain why bats with positive body swabs did not always have accompanying positive oral or rectal swabs and is consistent with the high frequency of positive oral swabs. It is possible that the nucleic acid detected on bat 7633EDB did not originate from that bat itself. In addition to that, isolation in PaKi cells may have failed due to low viral titer, incompatible cell type that lacked the receptor for this new virus, or perhaps the bat may have cleared the virus by the time we sampled urine for the isolation attempt. As fruit bat cells have homologs of furin proteases that are capable of cleaving/activating furin-dependent virus proteins, and there is a high level of conservation among *Pteropus alecto* furin and proteases from other mammals [37], we could expect that this would not limit virus isolation in PaKi cells. As shown in Figure 4, the virus was detected less frequently in swabs collected later in the longitudinal study, which may indicate that the virus was trending toward clearance by the end of the study. By evaluating both the broad virome patterns in a colony in a longitudinal fashion, as well as the patterns of distinct known and previously unknown viruses, we can continue to survey the ecological patterns of viral circulation and shedding by bat populations. It is important to evaluate both, as various studies have detected different patterns of paramyxovirus shedding [8,25,26].

With the full-length genome sequence, the virus could potentially be rescued by reverse genetics and the biology of the virus extensively characterized, as was recently accomplished for a novel bat paramyxovirus, myotis bat morbillivirus (MBaMV) [38]. Using the complete sequences for HN and F, expressing viral HN and F proteins to assess cross-reactivity in human sera or to evaluate bat sera for the presence of DbPV-binding antibodies may be useful. As our study was designed to be as least invasive as possible, we did not sacrifice any bats to obtain tissue samples to analyze. The pathogenesis and shedding pattern of DbPV is unknown, and body swabs may not be optimal for virus culture, highlighted by the variability of detection of DbPV in different swab types at different time points during this longitudinal study. Publicly available sequence data for bat-borne MuVs have been generated from spleen [1], kidney [39], oral, rectal, head, and body swabs [10]. Further investigation of similar sample types may expand the knowledge of the evolutionary relationship between bats and mammalian paramyxoviruses.

## Figures and Tables

**Figure 1 viruses-15-00659-f001:**
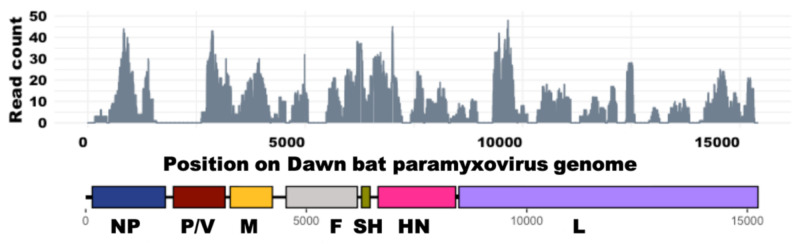
Coverage of deduplicated reads from bat 7633EDB body swab that mapped to DbPV scaffolded genome, with estimated genes depicted below the x-axis. Gaps between contigs exist where read count is 0.

**Figure 2 viruses-15-00659-f002:**
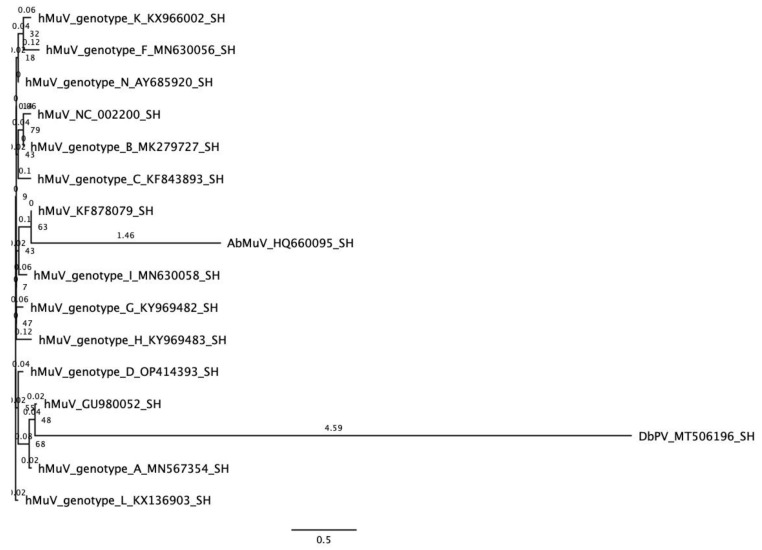
Maximum likelihood tree of complete SH gene product. While the nearest known neighbor to DbPV is AbMuV (HQ660095), DbPV does not cluster with AbMuV in phylogenetic analysis of the complete SH gene product. The unrooted tree is labeled with bootstrap values and branch lengths.

**Figure 3 viruses-15-00659-f003:**
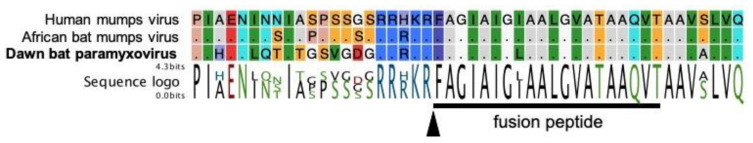
The conservation of a polybasic furin cleavage site among MuV (NC_002200), AbMuV (HQ660095), and DbPV in an alignment of amino acid sequences. Both AbMuV and MuV rely on proteolytic cleavage of the F protein for host cell binding and entry. Given the conservation of this furin cleavage site in DbPV, it is possible that this function is also required for the spread of infection and cell-to-cell fusion of the novel virus. Black arrowhead indicates cleavage site.

**Figure 4 viruses-15-00659-f004:**
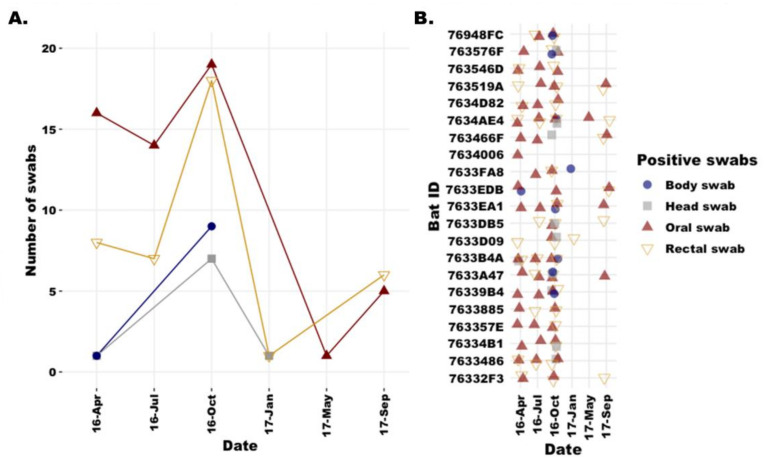
Frequency of detection among swab types and bats within the captive colony. (**A**) The number of swabs from which DbPV was detected in each swab type is graphed by sample collection date. The total number of head (grey square), body (navy circle), oral (red, solid triangle), and rectal (gold, hollow triangle) swab samples that contained reads mapping to DbPV are plotted by collection date. (**B**) Specific swab and bat ID information for each swab from which reads mapped to DbPV are plotted with individual bats on the y-axis and date on the x-axis. Color/shape corresponds to swab type.

**Table 1 viruses-15-00659-t001:** Prevalence of novel bat mumps virus at each time point.

Date	# Bats Sampled	# Bats Positive	% Bats Positive	# Swabs Sequenced	# Swabs Positive	% Swabs Positive
Apr-16	18	18	100.00%	41	26	63.41%
Jul-16	19	16	84.21%	28	21	75.00%
Oct-16	20	20	100.00%	75	53	70.67%
Jan-17	11	2	18.18%	14	2	14.29%
May-17	15	1	6.67%	21	1	4.76%
Sep-17	13	8	61.54%	27	11	40.74%

## Data Availability

The datasets supporting the conclusions of this article are available in the National Center for Biotechnology Information (NCBI) Sequence Read Archive (SRA), BioProject ID PRJNA561193 and under GenBank accession number MT506196.

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
