# Peer review of "Genomic Characterization of a Relative of Mumps Virus in Lesser Dawn Bats of Southeast Asia"

_viruses, 2023, doi:10.3390/v15030659_

Round 1

Reviewer 1 Report (Previous Reviewer 1)

The author's responses to my comments are acceptable and the quality of the manuscript has been significantly improved. 

I have some minor comments: 

1. Figure 2A: Numbers indicating the branch lengths in the tree are too small.

2. Lines 240: Please include the date the urine sample that was used for the virus isolation attempt was collected from bat 7633EDB

3. Line 317: Please include the time interval between the last positive swab collected from bat 7633EDB and the time the urine sample was collected that was used for the virus isolation attempt

4. Line 314: ".. whether the virus shed by bat 7633EDB itself....." : This should probable read: ".....whether the virus was shed by bat 7633EDB itself ....." 

Author Response

The author's responses to my comments are acceptable and the quality of the manuscript has been significantly improved. 

Thank you

I have some minor comments: 

  1. Figure 2A: Numbers indicating the branch lengths in the tree are too small.

A new tree has been generated with branch length and bootstrap labels in font size 20.

  1. Lines 240: Please include the date the urine sample that was used for the virus isolation attempt was collected from bat 7633EDB

The urine sample was collected from bat 7633EDB on 27 March 2019. This has been added to page 8.

  1. Line 317: Please include the time interval between the last positive swab collected from bat 7633EDB and the time the urine sample was collected that was used for the virus isolation attempt

The urine sample was collected from bat 7633EDB on 27 March 2019, 18 months after the last positive swab collected from bat 7633EDB. This has been added to page 8.

  1. Line 314: ".. whether the virus shed by bat 7633EDB itself....." : This should probable read: ".....whether the virus was shed by bat 7633EDB itself ....." 

This has been edited in the text.

Reviewer 2 Report (New Reviewer)

Review of resubmitted manuscript [#2184600] authored by Paskey et al, entitled Genomic characterization of a relative of mumps virus in lesser dawn bats in Southeast Asia. This manuscript has already received comments from 3 independent reviewers.

The revisions to the manuscript thoroughly address the reviewers' comments. The manuscript is now a comprehensive characterisation of the dawn bat paramyxovirus (DbPV genome). The authors have added more data (Table 1 and Figure 5) to clarify the detection frequency of DbPV in bats within the captive colony. Also, they have described methodological limitations in their discussion to better explain the lack of virus isolation they observed. I have two additional points that should be addressed by the authors prior to publication:

1.     In the abstract and discussion the authors state DbPV shares 74.8% nucleotide identity with AbMuV and 73.9% identity with MuV, however don’t explain how this has been calculated. Is this a genome-wide pairwise comparison and if so, is it valid/accurate with 25% of Ns and potentially shorter  HN and F genes? Perhaps amino acid identity of L (for which you have identified a complete sequence) is a more accurate statistic to include here. Overall, including a sentence acknowledging the limitations of a partial genome might be sage.

2.     Include bootstrap support values in Figure 2B

Following the inclusion of those two changes I think the manuscript is suitable for publication, however the authors may want to address the following at their discretion:

3.     A brief sentence about the ecology of wild E. spelaea in the introduction might set the scene for zoonotic transmission risk

4.     Do PaKi cells express furin. Perhaps a reference and comment on this as it could be one factor limiting virus isolation?

5.     E. spelaea probably groom each other. This could explain why you detect copious genome on one individual bat body swab but not in accompanying urine/fecal swabs. Also, this would be consistent with the high frequency of positive oral swabs observed.

Author Response

Review of resubmitted manuscript [#2184600] authored by Paskey et al, entitled Genomic characterization of a relative of mumps virus in lesser dawn bats in Southeast Asia. This manuscript has already received comments from 3 independent reviewers.

The revisions to the manuscript thoroughly address the reviewers' comments. The manuscript is now a comprehensive characterisation of the dawn bat paramyxovirus (DbPV genome). The authors have added more data (Table 1 and Figure 5) to clarify the detection frequency of DbPV in bats within the captive colony. Also, they have described methodological limitations in their discussion to better explain the lack of virus isolation they observed. I have two additional points that should be addressed by the authors prior to publication:

  1. In the abstract and discussion the authors state DbPV shares 74.8% nucleotide identity with AbMuV and 73.9% identity with MuV, however don’t explain how this has been calculated. Is this a genome-wide pairwise comparison and if so, is it valid/accurate with 25% of Ns and potentially shorter HN and F genes? Perhaps amino acid identity of L (for which you have identified a complete sequence) is a more accurate statistic to include here. Overall, including a sentence acknowledging the limitations of a partial genome might be sage.

As Reviewer 2 recommended, we have added more details to our methods and adjusted language throughout the text to better acknowledge the partial genome and provide clarity. We have also modified the statistic included to reflect the percent amino acid identity (as calculated by pairwise comparison, consistent with other amino acid-level comparisons described in the methods section) for a subset of L – the RNA-dependent RNA polymerase. This additional calculation (of RdRp) has been added to the methods section on page 3.

  1. Include bootstrap support values in Figure 2B

A new tree has been generated with both bootstrap values and branch lengths in font size 20.

Following the inclusion of those two changes I think the manuscript is suitable for publication, however the authors may want to address the following at their discretion:

  1. A brief sentence about the ecology of wild E. spelaea in the introduction might set the scene for zoonotic transmission risk

The suggested addition is now on page 1. Eonycteris spelaea are highly gregarious and known to co-roost in large populations of multiple bat species in caves across Southeast Asia [4].

  1. Do PaKi cells express furin. Perhaps a reference and comment on this as it could be one factor limiting virus isolation?

Thank you for sharing this idea. As fruit bat cells have homologs of furin proteases that are capable of cleaving/activating furin-dependent virus proteins, and there is a high level of conservation among Pteropus alecto furin and proteases from other mammals [2], we could expect that this would not limit virus isolation. This has been addressed in the text on page 10.

  1. E. spelaea probably groom each other. This could explain why you detect copious genome on one individual bat body swab but not in accompanying urine/fecal swabs. Also, this would be consistent with the high frequency of positive oral swabs observed.

Thank you, this germane point has been added to the discussion on page 10. The gregarious nature and grooming behavior of lesser dawn bats could explain why bats with positive body swabs did not always have accompanying positive oral or rectal swabs and is consistent with the high frequency of positive oral swabs. It is quite possible that the nucleic acid detected on bat 7633EDB did not originate from that bat itself.

Reviewer 3 Report (New Reviewer)

This manuscript written by Paskey et al. reported a novel genome of a mumps-like virus in lesser dawn bats collected in Southeast Asia. They tried to characterize the genomic features of the virus, however, only several contigs of the virus were obtained and flanked by “N” in the submitted GenBank file. Therefore, the reliability of these results in the study should be questioned, especially the genomic features and phylogenetic analyses.

Major comments:

The genomic sequence of the dawn bat paramyxovirus (DbPV), submitted by the authors (Accession NO.: MT506196) is 15172 bp in length, however, the sequence contains 3,603 “N”, accounting for 23.7% of the whole sequences. Moreover, this “N” is not only located in the non-coding regions but also scattered in the CDSs of the genome. For example, the Large protein which is 2,261 amino acids in length (Accession NO.: UDF54476), contains 594 “X amino acids” which translated from codons contains “N”. Therefore, the genomic comparison and phylogenetic tree may not reflect the actual evolutionary relationship of this virus with other viruses in the family Paramyxoviridae.

    Considering the bats could be infected by more than one paramyxovirus simultaneously, the submitted sequence might be a chimera. As all the downstream analyses in the paper were based on the sequence, the authors should obtain the complete genome, then relative analyses could be performed again.

Author Response

This manuscript written by Paskey et al. reported a novel genome of a mumps-like virus in lesser dawn bats collected in Southeast Asia. They tried to characterize the genomic features of the virus, however, only several contigs of the virus were obtained and flanked by “N” in the submitted GenBank file. Therefore, the reliability of these results in the study should be questioned, especially the genomic features and phylogenetic analyses.

Major comments:

The genomic sequence of the dawn bat paramyxovirus (DbPV), submitted by the authors (Accession NO.: MT506196) is 15172 bp in length, however, the sequence contains 3,603 “N”, accounting for 23.7% of the whole sequences. Moreover, this “N” is not only located in the non-coding regions but also scattered in the CDSs of the genome. For example, the Large protein which is 2,261 amino acids in length (Accession NO.: UDF54476), contains 594 “X amino acids” which translated from codons contains “N”. Therefore, the genomic comparison and phylogenetic tree may not reflect the actual evolutionary relationship of this virus with other viruses in the family Paramyxoviridae.

Reviewer 3 has expressed valid concern about the partially complete genome and the possibility that this novel virus could differ more from previously known mumps viruses than we were able to appreciate given the existing data. At this point in time, our understanding is that the virus is most closely related to another bat-borne mumps virus that was reported in 2012 [1] and comparisons at the amino acid level consistently recapitulate this assertion. Insights based on our existing genomic comparisons include the presence of the SH gene, which is not present in all orthorubulaviruses [3, 7] and informative for the purpose of primer design when selecting screening targets for environmental samples. As more sequences are deposited in public databases, as we have through this work, our understanding of the actual evolutionary relationship will continue to improve.

    Considering the bats could be infected by more than one paramyxovirus simultaneously, the submitted sequence might be a chimera. As all the downstream analyses in the paper were based on the sequence, the authors should obtain the complete genome, then relative analyses could be performed again.

We acknowledge the caveat that de novo assemblies and even extensive manual analysis as were utilized in our study could suffer from the inability to distinguish among similar but not identical viruses coinfecting one host. Given the recapitulation of these data using both assembly of reads by metaSPAdes [5], which is designed with the complexities of samples with multiple related strains in mind, and the classification of the joined reads using VirusSeeker [8] plus manual analysis, we hope to have avoided this pitfall. We hypothesize that low titer and sample quality (due to sequencing environmental swabs sampling the body of a bat) contribute to the breaks in the assembly, particularly because VirusSeeker classified the reads for DbPV in one subset that was distinct from other paramyxovirus sequences that were described in our longitudinal virome study [6] (these sequences were not discussed in this report because they were not mumps-like). Even in the case of successful isolation or rescue by reverse genetics and sequencing from culture, the caveat remains that in vitro passage could introduce culture-associated adaptation and result in differences in those sequence data, as well. As there are no remaining primary samples and we were unable to isolate the virus at a later sampling date, we believe that the composite sequence and our hypotheses should be shared with others within the community of bat-borne virus researchers who may come across similar sequences in the future.

References

  1. Drexler JF, Corman VM, Muller MA, Maganga GD, Vallo P, Binger T, Gloza-Rausch F, Cottontail VM, Rasche A, Yordanov S, Seebens A, Knornschild M, Oppong S, Adu Sarkodie Y, Pongombo C, Lukashev AN, Schmidt-Chanasit J, Stocker A, Carneiro AJ, Erbar S, Maisner A, Fronhoffs F, Buettner R, Kalko EK, Kruppa T, Franke CR, Kallies R, Yandoko ER, Herrler G, Reusken C, Hassanin A, Kruger DH, Matthee S, Ulrich RG, Leroy EM, Drosten C (2012) Bats host major mammalian paramyxoviruses. Nat Commun 3:796
  2. El Najjar F, Lampe L, Baker ML, Wang LF, Dutch RE (2015) Analysis of cathepsin and furin proteolytic enzymes involved in viral fusion protein activation in cells of the bat reservoir host. PLoS One 10:e0115736
  3. Johnson RI, Tachedjian M, Rowe B, Clayton BA, Layton R, Bergfeld J, Wang LF, Marsh GA (2018) Alston Virus, a Novel Paramyxovirus Isolated from Bats Causes Upper Respiratory Tract Infection in Experimentally Challenged Ferrets. Viruses 10
  4. Luo Y, Li B, Jiang RD, Hu BJ, Luo DS, Zhu GJ, Hu B, Liu HZ, Zhang YZ, Yang XL, Shi ZL (2018) Longitudinal Surveillance of Betacoronaviruses in Fruit Bats in Yunnan Province, China During 2009-2016. Virol Sin 33:87-95
  5. Nurk S, Meleshko D, Korobeynikov A, Pevzner PA (2017) metaSPAdes: a new versatile metagenomic assembler. Genome Res 27:824-834
  6. Paskey AC, Ng JHJ, Rice GK, Chia WN, Philipson CW, Foo RJH, Cer RZ, Long KA, Lueder MR, Frey KG, Hamilton T, Mendenhall IH, Smith GJ, Wang LF, Bishop-Lilly KA (2020) The temporal RNA virome patterns of a lesser dawn bat (Eonycteris spelaea) colony revealed by deep sequencing. Virus Evol 6:veaa017
  7. Wang LF, Hansson E, Yu M, Chua KB, Mathe N, Crameri G, Rima BK, Moreno-Lopez J, Eaton BT (2007) Full-length genome sequence and genetic relationship of two paramyxoviruses isolated from bat and pigs in the Americas. Arch Virol 152:1259-1271
  8. Zhao G, Wu G, Lim ES, Droit L, Krishnamurthy S, Barouch DH, Virgin HW, Wang D (2017) VirusSeeker, a computational pipeline for virus discovery and virome composition analysis. Virology 503:21-30

Round 2

Reviewer 3 Report (New Reviewer)

The authors have revised the manuscript finitely, yet the fundamental problem about the unverified partial viral sequence remains unresolved. The results of genome organization, sequence comparison and phylogenetic analyses were obtained based on unreliable data, involving Fig. 1, Fig. 2, and Fig. 3. Relevant results and conclusions were inaccurate or misleading.

Major comments:

1. The major fault of genome organization in this manuscript is the preconception that paramyxoviruses are conserved in structure, even if they belonged to distinct viral species and hosted by different mammalian species. For example, Beilong virus and J virus are closely related and both belong to the genus Jeilongvirus, however, the former possesses an X gene locating between G and L genes, while the latter has an extraordinary large G gene. Moreover, genomic organization variation had also been observed among different strains of Beilong virus: Beilong viruses in wild rats have one X gene, while the Beilong virus DQ100461 has two X genes (Patrick et al. 2016). The authors filled the scattered viral contigs by a huge number of “Ns” based on this preconception. Ironically, they then deduced the conclusion that the newly described Dawn bat paramyxovirus (DbPV), a novel viral species, has the same genome architecture with mumps virus

The genome organization of DbPV cannot be "predicted" based on the existing sequence, the corresponding results shown in Fig. 1 and Fig. 2A were misleading. As gaps accounted for 23.7% of the sequence and scattered in the whole genome (see my comment before), the positions and sizes of some ORFs of DbPV could not be determined. The genome organizations shown in Fig. 1 and Fig. 2A were actually imitated based on the sequence of the closest virus, not the real sequence of DbPV. For example, as shown in Fig. 1, no read was obtained around 2,500 bp, generating a long gap (~1,000 bp) and partial NP and P/V genes, it's unable to confirm the termination site of NP as well as the initiation site of P/V. Similar gaps scattered in the other genes, including F, HN, and L.

2. Another major fault about viral genome in this manuscript is the misconception that nucleotide and amino acid identity is consistent across the genes and genome. For example, the RdRp domain in the L protein exhibited the highest amino acid identity across genome between different viral species in paramyxoviruses, where historically were the target region for consensus PCR. In contrast, the remaining region of L protein exhibited lower identities. Therefore, the sequence identities throughout the manuscript were not true.

The sequence identities shown in Fig. 3 which were calculated based on the low-quality sequences containing gaps were unreliable.

3. Due to the large-scale sequence deficiency in the L gene (26.3% sequence loss which leading to seven gaps, see my comment before), the existing L amino acid sequence containing 594 "X amino acids" could NOT be used for relevant analyses directly. The phylogenetic tree in Fig. 2B constructed based on the low-quality sequence of L protein DbPV was inaccurate.

4. Throughout the manuscript, the sequences used for comparison, identity calculation, and phylogenetic analyses were described ambiguously. To evaluate the true evolutionary relationships, accurate descriptions of the sequences (partial or complete, the length) used for analyses are very important.

Reference:

Woo PCY, Wong AYP, Wong BHL, Lam CSF, Fan RYY, Lau SKP, Yuen KY. Comparative genome and evolutionary analysis of naturally occurring Beilong virus in brown and black rats. Infect Genet Evol. 2016; 45: 311-319.

Author Response

The authors have revised the manuscript finitely, yet the fundamental problem about the unverified partial viral sequence remains unresolved. The results of genome organization, sequence comparison and phylogenetic analyses were obtained based on unreliable data, involving Fig. 1, Fig. 2, and Fig. 3. Relevant results and conclusions were inaccurate or misleading.
In this revision, we have restructured the presentation of results by restricting analysis in Figure 2 to fully complete SH sequences as well as by adding/moving other supporting materials to supplemental. We have also edited accompanying text in response to the reviewer’s critiques with the goals of clarity and transparency such that readers may understand the limitations of the data and analyses presented in the manuscript. 
Over the past week or so an earnest attempt was made to assemble a more complete viral sequence by using several updated assembly pipelines (SPAdes and Unicycler), as well as by using a targeted filtering approach by which reads were mapped to all NCBI taxa corresponding to paramyxoviruses prior to assembly. The good news is that the existing results we report in this manuscript were confirmed and validated by the contigs obtained using these new approaches; however, unfortunately, a more complete genome did not result. We maintain that despite the known difficulties of detecting virus genomes within metagenomic samples, the contigs presented in this work cover a significant proportion or breadth of the anticipated size of the new genome. As incomplete viral genomes have been widely published, including in this same journal, Viruses, as recently as this week (in Novel Chaphamaparvovirus in Insectivorous Molossus molossus Bats, from the Brazilian Amazon Region published by Ramos et al., 2023), we believe that our work meets the current standard for sequencing-based surveillance and discovery in the field of bat-borne research.
Major comments:
1. The major fault of genome organization in this manuscript is the preconception that paramyxoviruses are conserved in structure, even if they belonged to distinct viral species and hosted by different mammalian species. For example, Beilong virus and J virus are closely related and both belong to the genus Jeilongvirus, however, the former possesses an X gene locating between G and L genes, while the latter has an extraordinary large G gene. Moreover, genomic organization variation had also been observed among different strains of Beilong virus: Beilong viruses in wild rats have one X gene, while the Beilong virus DQ100461 has two X genes (Patrick et al. 2016). The authors filled the scattered viral contigs by a huge number of “Ns” based on this preconception. Ironically, they then deduced the conclusion that the newly described Dawn bat paramyxovirus (DbPV), a novel viral species, has the same genome architecture with mumps virus
The genome organization of DbPV cannot be "predicted" based on the existing sequence, the corresponding results shown in Fig. 1 and Fig. 2A were misleading. As gaps accounted for 23.7% of the sequence and scattered in the whole genome (see my comment before), the positions and sizes of some ORFs of DbPV could not be determined. The genome organizations shown in Fig. 1 and Fig. 2A were actually imitated based on the sequence of the closest virus, not the real sequence of DbPV. For example, as shown in Fig. 1, no read was obtained around 2,500 bp, generating a long gap (~1,000 bp) and partial NP and P/V genes, it's unable to confirm the termination site of NP as well as the initiation site of P/V. Similar gaps scattered in the other genes, including F, HN, and L.
Thank you for these comments and for the recommended reference. After reading the reference that was provided and performing additional analyses as described above in response to the first comment, we have incorporated these critiques into the manuscript. While it is not possible to generate more data from these samples because they are exhausted, we appreciate that there are limitations to the conclusions that can be drawn using the existing data and that it is possible due to the incompleteness of the genome that we cannot rule out the possibility of a major structural difference such as the X genes in Beilong virus. This has been added as a caveat to the text on page 3.
We appreciate that our approach may not capture any anomalies in genetic structure, and while we believe that it is reasonable to expect that the genetic arrangement 3’-N-P-M-F-SH-HN-L-5’ would be similar out of necessity for the virus to function, we cannot rule out the possibility of an unknown extra gene. BlastX results for each contig (there is a contig hit to each gene) have been added to a supplemental table. To be as transparent as possible about the work that was done, and to avoid possibly leading to conclusions that are unsupportable, we have revised Figure 2 to include only SH in the phylogeny, for which the complete sequence is known, and to exclude the comparison of estimated genomic structure. We have chosen to include the estimated structure in Figure 1 as it is clear where read support is lacking and edited the text to include the idea that it is possible there may be deviations in the structure that are unknowable at the present time. A supplemental table describing the BlastX hits has been included to help readers understand the similarities between these sequences and known neighbors, particularly regarding where the contigs may lie on the complete genome.
2. Another major fault about viral genome in this manuscript is the misconception that nucleotide and amino acid identity is consistent across the genes and genome. For example, the RdRp domain in the L protein exhibited the highest amino acid identity across genome between different viral species in paramyxoviruses, where historically were the target region for consensus PCR. In contrast, the remaining region of L protein exhibited lower identities. Therefore, the sequence identities throughout the manuscript were not true.
The sequence identities shown in Fig. 3 which were calculated based on the low-quality sequences containing gaps were unreliable.
The reviewer is concerned that the pairwise identity calculations among MuV sequences is inaccurate due to the presence of Ns within the DbPV sequences. Additionally, due to the presence of Ns in many of the sequences in the public database, the possibility for inaccuracies may be amplified. We cannot control for the incompleteness of the public databases, though we believe that these calculations are a useful comparison among known sequences. Figure 3 shows that amino acid identity is not consistent across the genome. Thus, we propose to include the results that were in Figure 3 as a supplemental figure. Our rationale for doing so is to provide an illustration that the contigs representing the novel virus are apparently as divergent from AbMuV as from human MuVs, which we believe to be an interesting finding. We acknowledge that due to the incompleteness of genomes will result in imperfect results and that these comparisons are most useful for gaining a relative understanding to how the existing data compare to known, publicly shared sequences.
3. Due to the large-scale sequence deficiency in the L gene (26.3% sequence loss which leading to seven gaps, see my comment before), the existing L amino acid sequence containing 594 "X amino acids" could NOT be used for relevant analyses directly. The phylogenetic tree in Fig. 2B constructed based on the low-quality sequence of L protein DbPV was inaccurate.
To address the reviewer’s concern that a partially complete L sequence may lead to inaccuracies in the phylogenetic tree, Figure 2 has been revised to include only a phylogeny of the complete SH gene (used to define hMuV genotypes) and the L gene data is moved to supplemental materials.
4. Throughout the manuscript, the sequences used for comparison, identity calculation, and phylogenetic analyses were described ambiguously. To evaluate the true evolutionary relationships, accurate descriptions of the sequences (partial or complete, the length) used for analyses are very important.
Thank you for this feedback. We have gone through each section of the manuscript and included more detail to avoid ambiguous language in describing the work that was done and the sequences used (including completeness or containing gaps).

This manuscript is a resubmission of an earlier submission. The following is a list of the peer review reports and author responses from that submission.

Round 1

Reviewer 1 Report

In this manuscript the authors claim to report the detection of a new bat borne MuV in a captive colony of lesser dawn bat in Southeast Asia. The new virus was dubbed dawn bat paramyxovirus (DbPV) Sequences belonging to DbPV were detected in six swabs collected from a captive lesser dawn bat colony between 2016 and 2018. However, taken these sequences together, they only cover approximately 75% of the scaffolded DbPV reference genome with an average depth of 9.6x.  Comparison of the partial polymerase gene sequence of DbPV virus to previously known sequences demonstrates that it is more closely related to bat and human MuV as compared to paramyxoviruses of other species. The authors attempted to isolate full-length replicating DbPV from bat urine using PaKi cells. However, no paramyxovirus sequences could be detected by PCR.

General comments:

While the identification of a new bat borne MuV in lesser dawn bats in Southeast Asia is very intriguing and may have important implications for our understanding of the origins of human mumps viruses as well as with respect to the possibility of spill-over into the human population, the actual discovery of this new virus did not occur with this report, as the authors acknowledge. The authors reported the identification of this new Mumps virus already in their publication in Virus Evolution in 2020 (reference 9). Therefore, it seems to be misleading if the authors claim that this is the first report of a MuV-like virus in bats outside of Africa.

My major concern with this report is the fact that the authors neither succeeded in recovering a complete nucleotide sequence of the entire genome of this novel bat MuV, nor did they succeed in the isolation of the virus. While the failure to isolate novel bat viruses, such as the novel African bat MuV, can be explained to be due for several reasons, partially discussed in this report, more effort should have been made to elucidate the complete genome sequence of the novel virus. For instance, was the bat from which 75% of the sequence was recovered, sacrificed and tissue samples (including brains) harvested and analyzed for presence of DbPV full length virus genomes ?. It appears that this should have been possible given that this is a captive bat colony. Due to the lack of a full-length genome, no downstream wet- lab experiments can be conducted at this time except those described here in silico (detection of furin cleavage site in the F protein sequence). It would be interesting to express the viral HN and F proteins and assess cross-reactivity of human sera with these proteins. In addition, sera of the bats should be analyzed for presence of DbPV specific antibodies. Ultimately, a recombinant DbPV would need to be generated to be able to study the biology of this virus, including neurotoxic potential.  Thus, the reporting of only a partial sequence of this novel virus appears to be premature.

Specific comments:

1.      The authors do not specify whether the six positive samples were all obtained from the same bat or from different bats. Furthermore, if the sequences were obtained from several bats, and if there was an overlap of sequences, it should be stated if the nucleotide sequences originated from different bats were 100% identical.

2.      Did the authors attempt to amplify the missing nucleotide sequences using RT-nested PCR using primers flanking the missing sequences?

3.      In figure 2b, the authors compared the sequence of the DbPV with that of Mumps virus and other bat paramyxoviruses as well as of NDV. While the Genebank accession number for the Mumps virus sequence is provided (apparently corresponding to the Miyahara strain, subtype B), a comparison to all other mu mumps genotypes should have been included in the phylogenetic analysis. Furthermore, the African bat mumps virus (HQ660095) should be highlighted in the tree shown in figure 2B to be able to easier recognize it.

4.      The authors do not delineate in more detail the regions in the genome that were not covered by sequenced reads. In particular, the missing portions in the F and HN genes should be delineated in more detail.

5.      According to Figure 2A, there is a large insertion found in the DbPV HN gene, which is very intriguing. Unfortunately, the authors do not elaborate on this insertion and do not provide information on whether the sequence of this insert shows any relation to a known viral or cellular genes. Also, it is unclear whether this might be due to a sequencing artefact. This should be discussed.

Reviewer 2 Report

The manuscript „A relative of mumps virus reported for the first time in lesser dawn bats in Southeast Asia” from Paskey et al. described the detection of the genome of a new virus in bats. Due to the NGS assay used for the detection, the authors are able to give a sequence which is good covered and enable them to compare it in detail with related viruses of the same genus. By comparing the authors demonstrate conservation of a cleavage site that could be important for cell binding and entry. The manuscript is well written, is original, the figures support the conclusions, and the main conclusions are consistent.

However, I would advise some major revisions.

1.       Unfortunately, the virus is just a sequence and the authors attempts to gain an isolate were not successful. This aspect should be addressed more thoroughly by pointing out the limitations and possible reasons behind it.

a)       I did totally confused how many swabs the authors get from the colony. In the citated source they give more information but they have to state here at least the total number of animals and the total number of swabs. This would also allow the authors to discuss the dimension of six swabs to probable 210 swabs (number of Paskey et al Virus Evolution 2020) and the prevalence of the virus.

b)      The discussion of the isolation attempt is inadequate. Why have they chosen this bat individual (was one of the positive swabs from it?), why urine (when the RNA was gained from body swab and rectal swab)? When the cytopathic effect was not caused by paramyxoviruses, what was the reason? What should be done to improve future isolation attempts or which other tools need to get established before it will be successful.

c)       The authors discuss the problem that they have gotten only low numbers of reads of this virus and hypothesize that it is inconsistently shedded. However, as the virus was found in a captive colony which gives the possibility to have information about the infected animals. In future studies they could try to get more samples in shorter time intervals to confirm their hypothesis, in this manuscript at least some information about the animals associated with positive swabs should be included (even if the body swab does not automatically link the virus to a specific individual, the rectal swab does!). At least one animal died (according to Paskey et al virus evolution 2020), this or other dead individuals could be sectioned and scanned for the virus. If no dead individuals are available, clinical data about the animals should be included. Even no clinical signs (if they are adequate documented) give a hint that their implications of inapparent infections are true.

2.       The finding of a new virus should include a paragraph about the recent taxonomy according to the International Committee of Virus Taxonomy and what are the next putative relatives. The authors show a phylogeny with the next relatives, however, the two genera of Rubuluvirinae, Orthorubulavirus and Pararubulavirus, are not mentioned. If there are any rules of species demarcation of Rubuluvirinae, they should be applied and a proposal should be included if the new dawn bat paramyxovirus should be a new virus in the species Bat mumps orthorubulavirus or is a virus in a new species. If this proposal is not possible, a discussion should be included what information are missing to get a clear demarcation.

Minor revision: The low amino acid identity in the small hydrophophobic protein (SH) is outstanding in comparison to the other proteins. A short discussion, why higher variance in this protein can occur in regard to the function of the protein, would be helpful.

Reviewer 3 Report

By mining HTS data, Paskey et al. report the discovery of a new mumps virus in lesser dawn bats. The virus is 74.8% nt identical to its closest relative. They did not obtain the complete genome. The isolation using cell culture also failed.

Major concerns

1.      Please provide the results of RT-PCR detection in these samples, especially the bat urine (Bat ES10) used to inoculate cells. Mapping reads cannot give a solid result for the presence of the virus in certain samples.

2.      Please detail the six DbPV-positive swabs. The section Frequency of detection in the captive colony only describes four of them. Please also discuss the detection of DbPV RNA in six swabs.

3.      It’s advised to fill gaps using RT-PCR.

Minor concerns:

1.      Line 23: à bat-borne mumps viruses

2.      Lines 46: Rubulavirus should be in italic.

3.      Lines 175-180: Put the method details into the section M&M.

4.      Lines 187-189: This inference is meaningless; I recommend deleting it.

5.      Line 214: à it seems